# Polyurethane Foam Rafts Supported In Vitro Cultures of *Rindera graeca* Roots for Enhanced Production of Rinderol, Potent Proapoptotic Naphthoquinone Compound

**DOI:** 10.3390/ijms23010056

**Published:** 2021-12-21

**Authors:** Mateusz Kawka, Irena Bubko, Mirosława Koronkiewicz, Beata Gruber-Bzura, Konstantia Graikou, Ioanna Chinou, Małgorzata Jeziorek, Agnieszka Pietrosiuk, Katarzyna Sykłowska-Baranek

**Affiliations:** 1Department of Pharmaceutical Biology and Medicinal Plant Biotechnology, Faculty of Pharmacy, Medical University of Warsaw, 1 Banacha St., 02-097 Warsaw, Poland; mjeziorek@wum.edu.pl (M.J.); agnieszka.pietrosiuk@wum.edu.pl (A.P.); katarzyna.syklowska-baranek@wum.edu.pl (K.S.-B.); 2Department of Biochemistry and Biopharmaceuticals, National Medicines Institute, 30/34 Chełmska St., 00-725 Warsaw, Poland; i.bubko@nil.gov.pl (I.B.); b.gruber@nil.gov.pl (B.G.-B.); 3Department of Drug Biotechnology and Bioinformatics, National Medicines Institute, 30/34 Chełmska St., 00-725 Warsaw, Poland; m.koronkiewicz@nil.gov.pl; 4Lab of Pharmacognosy and Chemistry of Natural Products, Department of Pharmacy, National & Kapodistrian University of Athens, Zografou, 15771 Athens, Greece; kgraikou@pharm.uoa.gr (K.G.); ichinou@pharm.uoa.gr (I.C.)

**Keywords:** *Rindera graeca*, Boraginaceae, rinderol, immobilization, in situ extraction, apoptosis

## Abstract

Unique phytochemical profile of plants belonging to Boraginaceae family provides a prolific resource of lipophilic pigments from the group of naphthoquinone derivatives. To overcome low compound content, the major obstacle of plant-based production, immobilization of *Rindera graeca* roots in in vitro cultures was implemented for efficient production of rinderol, novel furanonaphthoquinone derivative with anticancer properties. Chromatographic procedures revealed rinderol presence in extracts of all investigated root lines, derived both from root biomass and post-culture medium. Unexpectedly, in the second stage of the experiment, rinderol production was ceased in control, unmodified culture systems. On the contrary, roots immobilized on PUF rafts uniformly and stably produced rinderol, and its highest amount was noted for transformed root lines after 42 days of cultivation (222.98 ± 10.47 µg/flask). PUF occurred to be the main place of compound accumulation. Moreover, investigation of rinderol biological activity revealed its fast-acting cell death induction in HeLa cervical cancer cells at relatively low concentrations. Presented results revealed successful application of *R. graeca* roots immobilization on PUF rafts for production and in situ product removal of rinderol, novel lipophilic furanonaphthoquinone with suggested proapoptotic activity.

## 1. Introduction

Plant secondary metabolites, along with other natural products, have gone a long evolutionary pathway resulting in remarkable diversity of structures, physicochemical properties, and bioactivities [1]. As compounds produced naturally through complex metabolic pathways of specific plant species, these metabolites found applicability in many industries making a particular contribution to a pharmacy [2]. Serving humanity for millennia as natural therapeutics, plant-derived molecules are constantly the significant feedstock in the manufacture of drugs and source of lead compounds in drug development research [3].

Considering an undisputed value of plant secondary metabolites, methods of their production are the field of ongoing development, which already led to the successful application of plant biomass in in vitro culture-based production. Plants introduction to strictly controlled environment allows to maintain bioprocesses without risks and drawbacks of ground cultivation, encompassing also environmental friendliness as usage of renewable plant biomass cultured in vitro avoids geographical, ecological, or political limitations. Moreover, techniques of in vitro plant tissue culture, through plentiful options of culture system modifications allow for enhancement of secondary metabolites productivity, in which naturally low levels primarily limit commercial implementation [4,5,6].

Among biotechnological techniques invented so far and aimed at improving the yield of plant secondary metabolites production, one of the most spectacular results was published for biomass immobilization [7]. However, the dominant research direction has been focused on the immobilization of plant cells, while plant organs, remained unexplored.

Polyurethane foam (PUF), one of the most commonly used polymeric materials [8], with its lightweight, spongy, and resilient structure, effectively absorbs an aqueous medium, therefore fulfilling basic conceptual requirements of root biomass immobilizing raft material for application in plant tissue in vitro culture systems. Moreover, recent advancements in comprehension of PUF physicochemical properties resulted in a possibility for its more predictable application as a solid-extractant. The highest affinity of PUF toward nonionic, lipophilic compounds with multiple reactive proton donors allows for selective extraction of metabolites [9,10]. In the context of the current experiment, PUF rafts can be treated as a multifunctional, bioengineering solution for simultaneous immobilization of plant root biomass and in situ solid-phase extraction of produced plant secondary metabolites in in vitro culture system.

Rinderol (Figure 1) compound investigated in the presented report, was for the first time identified in in vitro root cultures of *Cynoglossum columnae* Ten. (Boraginaceae) [11]. Preliminary bioactivity screening demonstrated its distinguishing, significant in vitro cytotoxicity against various human cancer cell lines. Rinderol has been also previously isolated from roots of *Rindera graeca* (A. DC.) Boiss. & Heldr. (Boraginaceae) cultivated in vitro and its structure was elucidated using spectral methods (^1^H/^13^C, ^1^H-^1^H and ^1^H-^13^C correlation NMR spectroscopy, ESI-MS) [12]. The structure was also identified as naphthofuranin A in collected in Kunming (China) *Onosma paniculata* (Bur. et Franch.) roots [13].

The current work for the first time describes the biotechnological approach which enables the production of rinderol in in vitro root cultures of *R. graeca*, an endemic plant occurring naturally in mountain territories of Greece [12]. Biotechnological method developed under conditions of the current study, allows producing rinderol via bioengineering technique applying PUF rafts for root immobilization. Finally, the first attempt to investigate rinderol cytotoxicity mechanism toward cancer cells was carried out.

## 2. Results and Discussion

### 2.1. HPLC DAD Analysis of R. graeca In Vitro Root Cultures

HPLC-DAD-UV-VIS analysis of investigated extracts obtained from PUF-rafts-immobilized *R. graeca* in vitro cultures exhibited the presence of rinderol with its characteristic spectrum [11] in post-culture medium and PUF rafts of all four root lines subjected to the current experiment (Figure 2, Figure 3 and Figure 4). Determination of rinderol presence in extracts from root biomass revealed a lack of a significant amount of compound accumulated in plant tissue. Appendix A contain comprehensive overview of representative chromatograms from each root line (Appendix A).

### 2.2. Effect of Root Immobilization on Biomass Growth

Biomass growth of immobilized on PUFs *R. graeca* roots was evaluated in comparison to unmodified control cultures. The estimation of root growth was performed based on dry biomass accumulation. In control and immobilized cultures, root growth was observed till the end of culture, that is 42nd day with exception of the RgTR17 root line (Figure 5). The highest root biomass accumulation was achieved by root line regenerated from callus (RgCR) after 42 days of growth in the control culture (0.595 ± 0.070 g DW/flask; 17.45 ± 1.17-fold growth rate). When the immobilization procedure was accomplished, a significant but heterogeneous decrease in biomass accumulation was noted and the final dry weight, even up to 6.03-fold lower as compared to control was achieved (depending on the root line and the day of culture). In the case of immobilized roots, the highest increase of biomass was achieved by the line of anatomical roots (RgAR) after 42 days of cultivation (0.206 ± 0.030 g DW/flask; 5.72 ± 0.20-fold growth rate). Analysis of root growth data after 35 and 42 days of cultivation reveals significant differences between control and immobilized roots for respective root line and day of culture (*p* ≤ 0.05), moreover in cultures of transformed root line 17 (RgTR17) immobilized on PUF, their maximal biomass accumulation was observed after 35 days of culture.

The reported strong inhibition of biomass growth after root immobilization on PUF rafts could be attributed to the limited access of cultured roots to the medium nutrients or/and their deprivation due to nutrients absorption into PUFs. Roots placed on the PUF raft floating on the surface of the culture medium had restricted contact with the medium and were rather frequently lavaged together with movements of rotary shaker than submerged in the medium.

Previous investigations concerning the application of support materials in plant cell and tissue in vitro cultures do not provide much room for direct comparison with the presented study, considering various types of culture vessel (shake flask or bioreactor), biomass type (cells, roots or shoots), plant species, and supporting material used. The concept of plant cells immobilization has been already investigated multiple times, mostly with the application of calcium alginate for cells entrapment, however, PUF was also applied. Both materials used as a support for immobilization demonstrated mainly detrimental effects with respect to cell biomass growth [14,15,16]. However, immobilized plant cells were found to invade and remain inside porous polymeric materials, which differentiate them from root tissue that maintains only superficial contact with carrier. Application of PUF as support for plant roots has shown previously the lack of biomass growth-inhibiting effect [17] and superiority over some other similar materials in bioreactor systems [18]. Up to date, there have been no reports demonstrating a similar evident drop in biomass accumulation after plant root immobilization in in vitro cultures as was demonstrated in the current study.

### 2.3. Effect of Root Immobilization on Rinderol Production

*R. graeca* immobilization of roots upon PUF rafts resulted in the reduced intensity of pigment excretion into the medium which became more translucent than in unmodified control. The effect was accompanied by a deep, red coloring of PUF rafts caused by the sorption of pigment compounds produced from *R. graeca* roots (Figure 6).

Chromatographic analysis showed that irrespective of the root line, in control unmodified cultures rinderol was not detected in both root biomass and post-culture medium. Observed results were highly unexpected due to preliminary investigations and identification of significant amounts of rinderol in *R. graeca* root in vitro unmodified culture systems [19]. This phenomenon could be explained by somaclonal, temporary, or permanent variability that is a consequence of in vitro conditions and among them, an increasing number of subcultures [20].

Concerning the results from control cultures, root immobilization on PUF rafts efficiently stimulated rinderol intracellular productivity. Roots of each line produced rinderol in PUF rafts-supported culture systems (Figure 7). The highest productivity per flask was observed in roots of two transformed lines RgTR7 and RgTR17, with the maximum total yield noted on the 42nd day of cultivation, 222.98 ± 10.47 µg/flask and 169.75 ± 13.52 µg/flask, respectively. The lowest rinderol productivity per flask at the end of the 42nd day of culture was determined for root line RgCR and reached 105 ± 19.77 µg/flask. For all studied root lines, a continuous increase in rinderol production along cultivation time was observed. Analysis of productivity expressed per gram of dry weight leads to similar observations for each of investigated root lines. The highest yield of rinderol was noted for the RgTR7 root line, however, the maximum value was achieved on the 28th day of culture (3.77 ± 0.48 mg/g DW) with further continuous decrease which could be attributed to the reduction in root dry biomass accumulation (Figure 8).

More in-depth analysis of the shares of each of the culture system phases (roots biomass, culture medium and PUF rafts) demonstrated that PUF rafts not only induced rinderol biosynthesis but also almost entirely accumulated this compound. Observed results indicate a high affinity of produced lipophilic pigment rinderol toward PUF. As it has been recently described in the literature, PUF behaves as a selective absorption sink for lipophilic compounds in the aqueous medium [9].

Application of PUF as support for root tissue in in vitro culture systems, due to its multidimensional influence on cultivated biomass, should be discussed with consideration to particular constituent effects. First, the immobilization of plant cells or tissues with various support materials has been proved to enhance and stabilize the biosynthesis of secondary metabolites [21,22,23]. It is recognized that the rate of secondary metabolism in plants accelerates in the stationary phase of growth, therefore previously discussed growth inhibition caused by biomass immobilization might be an inherent feature of the successful procedure [21,24]. There are although examples demonstrating beneficial effects of root immobilization on PUF rafts for both, biomass growth and secondary metabolites productivity in bioreactor cultures [18]. Second, it was demonstrated that biomass immobilization enhanced the excretion of secondary metabolites like ajmalicine, scopolin, shikonin, or plumbagin into the culture medium, facilitating the isolation procedure of the produced compounds [25,26,27,28]. Referring to the presented results, the lack of rinderol in cultures without PUF rafts may indicate that PUF intensified this compound exudation from *R. graeca* in in vitro roots. The third aspect of investigated roots immobilization on PUF, the most distinguishing from previously published studies, is simultaneous functionality of examined material as an in situ adsorbent in in vitro culture system. In the presented experiment, PUF showed high efficiency in the induction of rinderol production with further accumulation of lipophilic rinderol excreted fully from root tissue. This strategy is coherent with the group of in situ product removal (ISPR) methods. ISPR has already a proven status in the enhancement of secondary metabolites production via plant in vitro cultures, although for the first time PUF was evaluated for such purpose. Numerous studies indicate that ISPR develops its full functionality in the production of metabolites which are unstable, vulnerable to intracellular degradation, or with productivity limited by feedback inhibition [6,29,30]. Considering complexity of interaction of plant biomass with PUF in the examined culture system, the possible mechanism of rinderol productivity enhancement would depend on modification of root tissue microenvironment with nutrients deprivation, induction of stationary growth-phase with secondary metabolism, and in situ adsorption of produced and excreted lipophilic compounds. In the context of the presented results, PUF rafts occurred to be the multifunctional solution for simultaneous immobilization of *R. graeca* roots and in situ product removal, leading to enhancement of rinderol yield.

### 2.4. Biological Activity of Rinderol

Previously reported considerable cytotoxic potential of rinderol demonstrated by Jeziorek et al. [11], creates the basis for investigations of rinderol mechanism of action. In the current study, the possible pathway of cell death, necrotic or apoptotic, in HeLa cervical cancer cell line was examined.

The highest cytotoxicity of rinderol against HeLa cells, measured in MTT assay as a 50% decrease in cell viability, was noted for 48 and 72 h exposure period, reaching IC_50_ values for 48 h and 72 h of 5.13 ± 0.67 µg/mL and 5.16 ± 0.28 µg/mL, respectively (Figure 9). The lowest cytotoxic activity of rinderol was observed after 24 h of incubation resulting in IC_50_ value of 5.66 ± 0.12 µg/mL. Obtained values are consistent with previous reports [11]. The results of the current study exhibited that rinderol caused continuous reduction in HeLa cells viability during 48 h of incubation with statistically significant change for a prolonged 72 h period.

To determine if rinderol induced cell death occurs via the apoptotic pathway, the flow cytometric assessment was performed by assessing the phosphatidylserine externalization in HeLa cells using Annexin V (FITC) staining concomitantly with propidium iodide to assess plasma membrane integrity. Analysis of obtained results suggests the time and dose-dependent cell death-inducing effect of rinderol (Figure 10). Interestingly, the investigated compound proved to be an outstandingly fast-acting agent. Statistically significant change was already observed after the first 4 h time point of exposure to IC_50_ (5.66 µg/mL) and doubled IC_50_ (11.32 µg/mL) concentration value for 24 h incubation time, which have induced apoptosis (respectively, 41.60 ± 3.72% and 76.63 ± 6.27%).

Fraction of cells in the apoptotic state (mainly late apoptotic, both annexin V and propidium iodide positive) amounted to 50% of the cell population before reaching 8 h of incubation with rinderol in 24 h IC_50_ concentration (5.66 µg/mL) (Figure 11). After 24 h of exposure in the same conditions, the percent of cells exhibiting phosphatidylserine externalization progressed to 90.03 ± 6.64%, showing a larger than expected magnitude of investigated proapoptotic effect (in reference to results of cell viability assay). No statistically significant change in the apoptotic fraction was observed for half of IC_50_ concentration until the 24 h time point of incubation. Analysis of cytograms (Figure 12) demonstrates the characteristic shift of HeLa cells state from healthy cells (lower left quadrant), through early apoptotic cells (lower right quadrant) to late apoptotic cells (upper right quadrant).

Analysis of alterations in ΔΨ_m_ shows that exposure of HeLa cells to rinderol at studied concentrations led to time and dose-dependent depolarization of mitochondrial inner membrane, which was observed as a decrease in cell population emitting red fluorescence (from J-aggregates) in favor of green (from JC-1 monomers) and the magnitude of this shift was increasing with rising rinderol concentration and incubation duration (Figure 13). The timing and magnitude of the depolarization effect were comparable with the above described characteristic changes in the plasma membrane (Annexin V-FITC/PI double staining). Until the 8 h time point of cells exposure to investigated compound at IC_50_ concentration for 24 h (5.66 µg/mL), almost half of HeLa cells population presented depolarization of the mitochondrial membrane. Analysis of cytograms from JC-1 assay (Figure 14) showed specific shift representing alteration of red aggregates to green monomers caused by the loss of mitochondrial membrane function.

Fluorescence imaging of HeLa cells stained with DAPI was applied for the detection of cell death. Performed morphological analysis of nuclear alterations caused by exposure to rinderol, revealed hallmarks of occurring apoptosis recognized as shrinkage of cytoplasm and nucleus, chromatin condensation and generation of apoptotic bodies. After 24 h of exposure to rinderol intensive progression of HeLa cells detachment was observed. Cells treated only with DMSO preserved the homogenous distribution of DAPI staining inside nuclei for the whole duration of the experiment (Figure 15).

Results obtained from presented investigation suggest dominant occurrence of programmed cell death mechanism induced by exposure of HeLa cells to rinderol. Observed decrease in mitochondrial membrane potential correlated well with externalization of phosphatidylserine outside cell membrane (Pearson’s correlation coefficient r = 0.92). Presented data are consistent with repeatedly reported involvement of mitochondrial pathway in the mechanism of apoptosis activation induced by naphthoquinones structurally similar to rinderol [31]. According to literature, commonly demonstrated intracellular events caused by shikonin derivatives and furanonapthoquinones comprise stress inducing generation of reactive oxygen species (ROS) with mitochondrial permeabilization [32,33,34,35,36] and activation of proapoptotic or pronecroptotic proteins [37,38]. Interestingly, it has been recently discovered for two currently clinically evaluated plant-derived naphthoquinones, β-lapachone and napabucasin, that magnitude of their ROS generating activity was dependent mainly on bioactivation by cytosolic NAD(P)H:quinone oxidoreductase 1 (NQO1) [39,40]. NQO1 overexpression can be found in multiple cancer cell lines being also considered as a poor clinical prognostic factor [41,42,43]. This distinguishing feature of cancer cells is one of the many reasons for dedicated interest into naphthoquinone derivatives as potentially more selective anticancer therapeutics. Furthermore, it was lately reported that angularly annellated isomers of furanonaphthoquinone napabucasin, the first-in-class cancer stemness inhibitor, demonstrate significantly higher anticancer potency [44]. Considering that rinderol also possess such preferential structure, further research would be highly justified.

## 3. Materials and Methods

### 3.1. Plant Material

Four root lines of *R. graeca*, developed via biotechnological methods as described by Sykłowska-Baranek et al. [45,46], were subjected to current experiments. Each of the four root lines were of different origin: (i) anatomical root line—RgAR; (ii) line of roots regenerated from callus—RgCR; (iii) transformed root line 7—RgTR7; (iv) transformed root line 17—RgTR17. During the whole experiment, roots of each line were cultivated in 250 mL Erlenmeyer flasks containing 50 mL of liquid hormone-free DCR medium [47]. All root cultures were carried out on a gyratory shaker (105 rpm, INFORS AG, Bottmingen, Switzerland) and maintained in 24 °C, in darkness. Routine subculturing was performed every 4 weeks.

### 3.2. Root Immobilization

Before sterilization of culture medium, rafts from polyurethane foam (PUF) (cuboids, approx. 30 mm × 30 mm × 10 mm) were placed inside the flask on the surface of aqueous phase. *R. graeca* roots cultivated for 28-days from the last subculture were used as an inoculum (0.20–0.25 g) and were placed on the surface of polyurethane foam raft floating in culture medium. Then, for the scheduled time period (4, 5 or 6 weeks) culture systems were incubated on a gyratory shaker (105 rpm, INFORS AG, Bottmingen, Switzerland) in 24 °C, in darkness.

### 3.3. Extraction Procedure

At set time points, that is on day 28th, 35th, and 42nd of culture, the roots were harvested, gently pressed on the paper filter, and weighted. Moreover, post-culture medium and PUF rafts were collected and stored at −20 °C. Prior to extraction procedures, plant material and PUF rafts were lyophilized (Christ ALPHA 1-4 LSC; Osterode am Harz, Germany). Growth of roots was measured and expressed as dry weight (DW) per flask [g DW/flask], while the growth rate was expressed as a ratio of final dry weight to initial dry weight. All solvents used for following extraction procedures were obtained commercially from Sigma-Aldrich (St. Louis, MO, USA). Dried and micronized roots and post-culture medium were extracted with n-hexane, while for cut pieces of PUF rafts methanol extraction was performed. The root material as well as PUF pieces were extracted by 15 min sonication (Sonorex; Bandelin, Berlin, Germany). Generally, the extraction was carried out till the solvent color fades. Extraction procedures from root biomass, post-culture medium, and PUF-rafts were followed by solvent removal on rotary evaporator (40 °C, Heidolph; Schwabach, Germany). Solid residue was collected and transferred to 2 mL plastic tubes with pure methanol (HPLC grade; Sigma-Aldrich, St. Louis, MO, USA) and the final drying of extracts was performed in centrifugal vacuum concentrator connected with cold trap (Heto Holten CT110; Heto Lab Equipment, Allerød, Denmark).

### 3.4. Isolation of Rinderol and HPLC-DAD-UV-Vis Analysis

Extracts obtained from all types of conducted *R. graeca* root cultures were collected and combined for preparative isolation with further purification of rinderol. All solvents used for following isolation procedures were purchased from Sigma-Aldrich (St. Louis, MO, USA). In the first step, combined extract (2.28 g) was subjected to preliminary vacuum liquid chromatography (VLC) separation (bed 6 × 5 cm, 60H silica gel sorbent, Merck, Darmstadt, Germany) for preliminary purification of investigated compound. VLC was conducted over 60H silica gel sorbent in gradient elution (A: n-hexane, B: dichloromethane, C: ethyl acetate; 200 mL, A: 100%; 200 mL, A: 50%, B: 50%; 200 mL, B: 100%; 200 mL, B: 50%, C: 50%; 200 mL, B:25%, C: 75%). Orange colored fractions containing rinderol were eluted with a mixture of methylene chloride:ethyl acetate (25:75) and further identified with HPLC-DAD-UV-Vis technique, combined and followed by solvent mixture removal on a rotary evaporator (40 °C, Heidolph; Schwabach, Germany). The second step of purification was fulfilled by separation on TLC plates (Merck Kieselgel 60 F_254_, 0.2 mm layer thickness; Darmstadt, Germany) with a mixture of n-hexane:ethyl acetate:acetic acid (70:29:1) as a mobile phase. Ethyl acetate was used to extract rinderol from compound zones of resulted TLC plates yielding rinderol (6.39 mg).

To prepare samples for HPLC-DAD-UV-Vis analysis, dried extracts from root biomass, post-culture medium, and PUF-rafts were dissolved in pure methanol (HPLC grade, Merck, Darmstadt, Germany). Chromatographic procedure (RP-HPLC) and further analysis were carried out using DIONEX HPLC system (Sunnyvale, CA, USA) connected with an automated sample injector (ASI-100) and UVD 340S UV-Vis diode-array detector, under following conditions: gradient elution—acetonitrile (60–80%)/0.04 M orthophosphoric acid (40–20%); flow rate—1.5 mL/min; run time—15 min; injection volume—20 µL; column: EC Nucleosil 120-7 ODS (250 × 4.6 mm, 7 μm particles, 120 Å pores, Macherey-Nagel, Allentown, PA, USA). Whole analysis was performed at room temperature. Eluent absorbance was monitored at 215, 237, 350, and 436 nm. The amount of rinderol in samples was calculated according to 237 nm wavelength. Rinderol standard of previously confirmed identity [11] was used for peak identification and calibration curve preparation.

### 3.5. Cancer Cell Line Culture

Rinderol bioactivity in vitro assays included monolayer cultures of human cervical cancer HeLa cell line. Cells were purchased from American Type Culture Collection (Manassas, VA, USA). Minimal Essential Medium (Gibco™, Thermo Fisher Scientific, Waltham, MA, USA) used for in vitro cell cultures was supplemented with 10% fetal bovine serum (Gibco™, Thermo Fisher Scientific, Waltham, MA, USA), 100 units/mL penicillin, 100 µg/mL streptomycin, 0.25 µg/mL amphotericin B (Gibco™ Antibiotic-Antimycotic, Thermo Fisher Scientific, Waltham, MA, USA). Cultures conditions were maintained in a humidified atmosphere with 5% CO_2_ and a temperature of 37 °C in an incubator.

### 3.6. Cytotoxicity Assay

Determination of rinderol cytotoxic activity was performed with the use of MTT assay [48,49,50]. HeLa cells were seeded on 96-well plates at a density of 5.6 × 10^3^ cells/well. After 24 h of incubation, cells were exposed to rinderol (in range of concentrations: 0–8 µg/mL) for further 24, 48, and 72 h of culturing. After the scheduled time of incubation with investigated compound, culture medium was removed and 50 µL of 5 mg/mL MTT salt (3-(4,5-dimethylthiazol-2-yl)-2,5-diphenyl tetrazolium bromide, Sigma-Aldrich, St. Louis, MO, USA) solution in PBS (phosphate buffered saline; Gibco™, Thermo Fisher Scientific Waltham, MA, USA) was added to each well for following 3-h incubation. In the next step, MTT solution was discarded and formed formazan crystals were dissolved by addition of 100 µL DMSO (Sigma-Aldrich, St. Louis, MO, USA) to each well and thorough mixing. Absorbance measurements were acquired using a plate reader (Victor 3 Plate Reader; Perkin Elmer, Waltham, MA, USA) at 570 nm and a reference wavelength of 620 nm. The experiment was performed in three independent trials. IC_50_ values were calculated from the Hill equation [51].

### 3.7. Flow Cytometry

Flow cytometry analyses were run on a FACSCanto II flow cytometer (BD Biosciences, San Diego, CA, USA) and analyzed using BD FACSDiva software.

### 3.8. Apoptosis Assay by Annexin V/Propidium Iodide (PI)

Apoptosis was measured using the FITC Annexin V Apoptosis Detection Kit I (BD Pharmingen, San Diego, CA, USA). HeLa cells were seeded on 6-well plates at a density of 2 × 10^5^ cells/well. After 24 h of incubation, cells were exposed to rinderol various concentrations (selected according to results of 24 h cytotoxicity assay: half of the IC_50_ = 2.83 µg/mL; IC_50_ = 5.66 µg/mL; doubled IC_50_ = 11.32 µg/mL) for 4, 8, 16, and 24 h incubation periods. After each of the investigated time points cells suspended in culture medium were collected and combined with cells detached by treatment with Accutase Cell Detachment Solution (BD Pharmingen, San Diego, CA, USA). Staining of cells with Annexin V (FITC) and propidium iodide (PI) was performed based on the manufacturer’s instructions. Flow cytometry measurements of cells were performed within 1 h after labeling. For all assays, number of gated events was set to 5000. At least three independent trials were performed for each experiment.

### 3.9. Mitochondrial Membrane Potential (ΔΨm) Assay

Alterations of mitochondrial membrane potential (ΔΨm) in HeLa cells induced by exposure to rinderol were investigated using JC-1 cationic dye (BD™ MitoScreen (JC-1); BD Pharmingen, San Diego, CA, USA). Cells were cultured on 6-well plates after inoculation at a density of 2 × 10^5^ cells/well. After 24 h of incubation cells were treated with rinderol in concentrations and for time periods analogical to apoptosis detection assay. Staining was performed according to the manufacturer’s instructions and then examined by flow cytometry. Samples used as positive control were incubated with 10 µg/mL of carbonyl cyanide m-chlorophenyl hydrazone (CCCP; Sigma-Aldrich, St. Louis, MO, USA). Results were presented as the percent of cell population with depolarized mitochondrial membrane. For all assays, the number of gated events was set to 10,000. At least three independent trials were performed for each experiment.

### 3.10. Morphological Analysis of Cells Nuclei

Morphological changes in cell nuclei after the exposure of HeLa cells to rinderol were examined using DAPI staining. HeLa cells were seeded on 6-well plates at a density of 2 × 10^5^ cells/well. After 24 h of incubation, cells were treated with rinderol with various concentrations (2.83 µg/mL; 5.66 µg/mL; 11.32 µg/mL) for incubation periods from 4 h to 72 h. After the scheduled time of exposure, cells were washed with PBS, and subjected to staining with 1.0 µg/mL DAPI solution (Sigma-Aldrich, St. Louis, MO, USA). Examination of morphological nuclei changes was performed using ZOE™ Fluorescence Cell Imager (Bio-Rad, Hercules, CA, USA). Images were analyzed in search of nuclear chromatin condensation and following fragmentation with formation of apoptotic bodies, features characteristic for apoptotic cell death.

### 3.11. Statistical Analysis

For analysis of growth of root biomass, rinderol productivity, cytotoxicity assessment, and flow cytometric experiments, one-way analysis of variance (ANOVA) was applied to evaluate the statistical difference between mean values. Normal distribution and variance homogeneity were examined with Shapiro–Wilk test and Bartlett’s test, respectively. For all experiments, *p* ≤ 0.05 was considered significant.

## 4. Conclusions

In the presented study the *R. graeca* root cultures were elaborated as an effective source of rinderol, a rare lipophilic naphthoquinone derivative which in previous studies exhibited considerable cytotoxic activity. Investigating biotechnological approaches for sustainable production of rinderol, stable restoration of its biosynthesis was observed after root immobilization on PUF rafts, in comparison to unmodified in vitro culture system. Elaborated biotechnological technique occurred to be effective in induction of rinderol production and in situ product removal by PUF rafts. At the same time PUF proved to be the main accumulation container of the target compound in investigated in vitro culture system. Finally, elucidation of rinderol biological activity mechanism in HeLa cervical cancer cells showed its relatively fast-acting cell death induction with suggested apoptotic mechanism. Presented results justify further efforts for more comprehensive evaluation of rinderol anticancer potential.

## Figures and Tables

**Figure 1 ijms-23-00056-f001:**
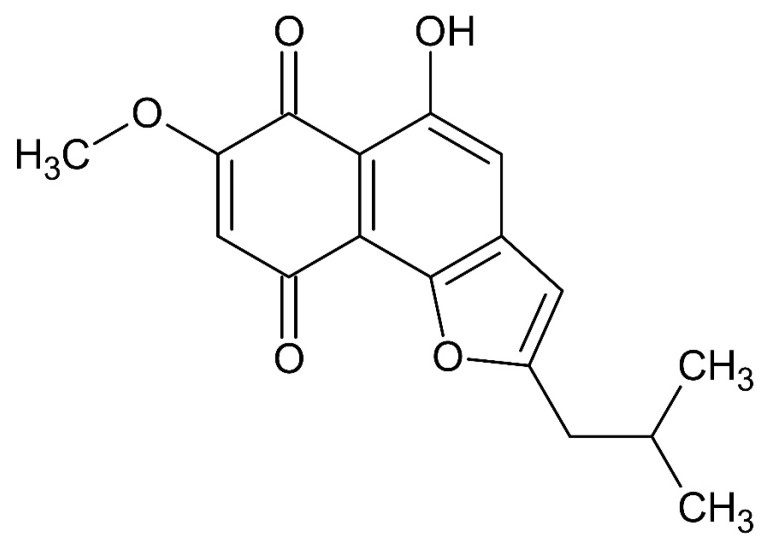
Chemical structure of rinderol.

**Figure 2 ijms-23-00056-f002:**
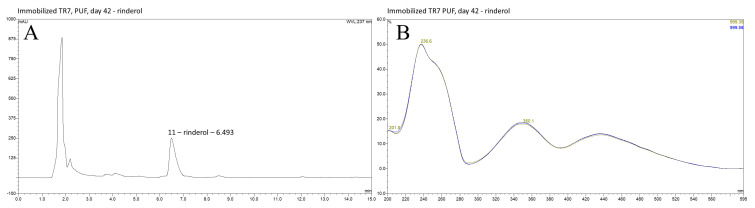
(**A**) HPLC DAD chromatogram of PUF methanol extract from the most productive *R. graeca* root line TR7 cultivated in vitro in PUF supported cultures; (**B**) Rinderol specific spectrum of peak 11 confirming compound identification, λ = 237 nm.

**Figure 3 ijms-23-00056-f003:**
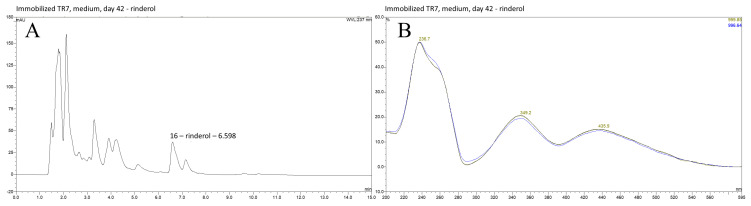
(**A**) HPLC DAD chromatograms of n-hexane extract from the post-culture medium of the most productive *R. graeca* root line TR7 cultivated in vitro in PUF immobilized cultures; (**B**) rinderol specific spectrum of peak 16 confirming compound identification, λ = 237 nm.

**Figure 4 ijms-23-00056-f004:**
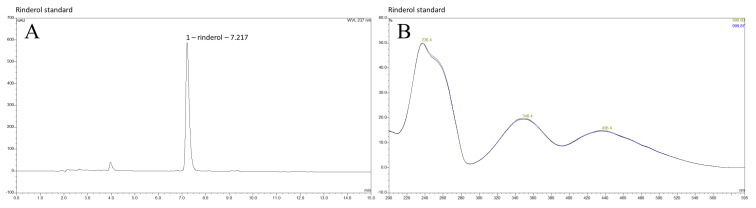
(**A**) Chromatogram of rinderol standard with the confirmed chemical structure used for compound identification in extracts obtained from investigated in vitro culture systems of *R. graeca* root lines; (**B**) UV-Vis spectrum of rinderol standard, λ = 237 nm.

**Figure 5 ijms-23-00056-f005:**
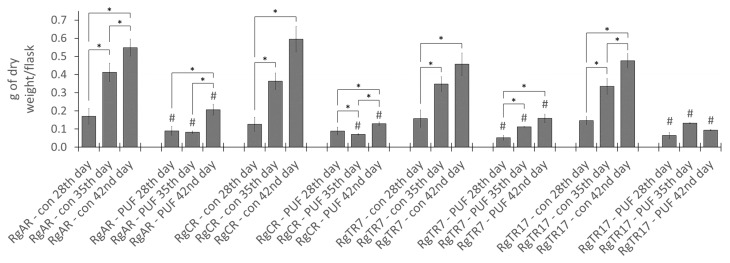
Dry weight biomass accumulation in cultures of four investigated *R. graeca* root lines after 28, 35, and 42 days of culturing with and without immobilization on PUF rafts. Presented data are the means ± SD of three independent replicates. Grids (#) indicate statistically significant differences (*p* ≤ 0.05) between treatments for respective time points within root line. Asterisks (*) above square brackets point to statistically significant differences between consecutive time points within the treatment and root line (*p* ≤ 0.05); “con”—control cultures of respective root lines without immobilization, “PUF”—cultures of respective root lines immobilized on PUF rafts.

**Figure 6 ijms-23-00056-f006:**
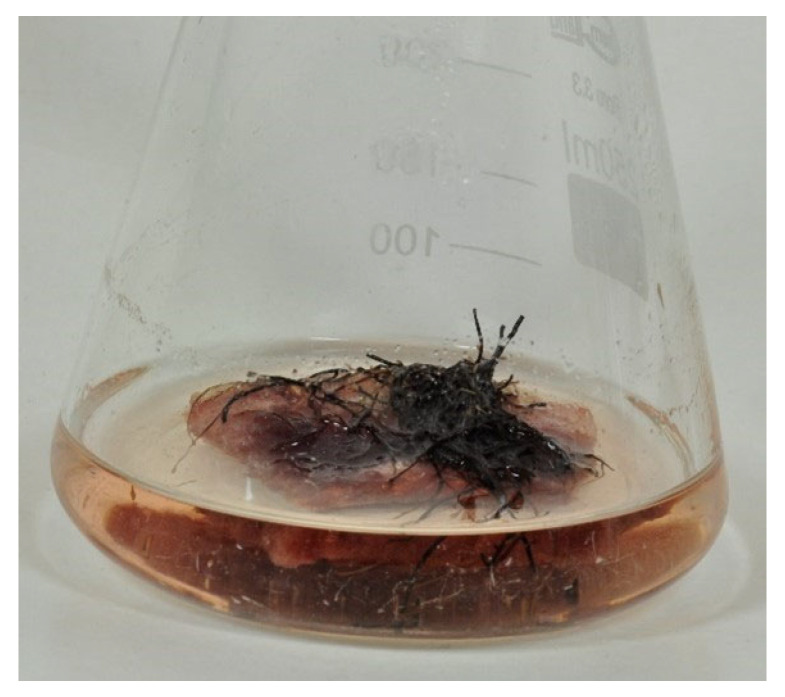
28th day in vitro culture of *R. graeca* roots immobilized on polyurethane foam raft.

**Figure 7 ijms-23-00056-f007:**
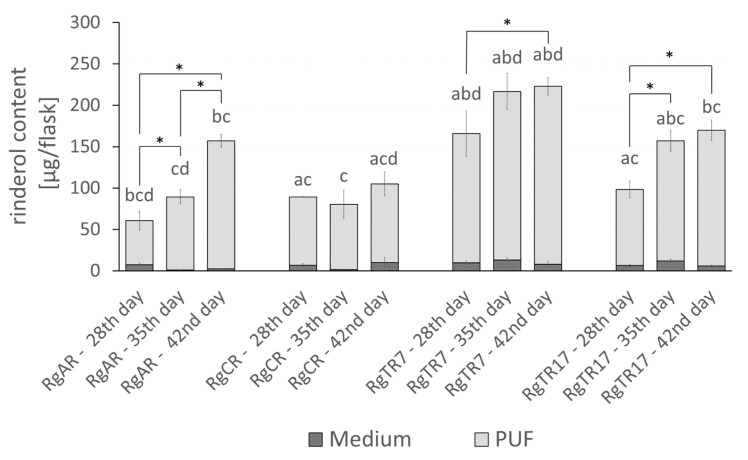
Rinderol productivity [µg/flask] in culture systems of *R. graeca* roots immobilized on PUF. Rinderol was detected only in post-culture medium and PUF, while it was not present in the root biomass. Presented data are the means ± SD of three independent replicates. Asterisks (*) above square brackets indicate statistically significant differences (*p* ≤ 0.05) in rinderol productivity between consecutive time points within each root line. Letters indicate statistically significant differences (*p* ≤ 0.05) in rinderol productivity between root lines: RgAR (a), RgCR (b), RgTR7 (c), and RgTR17 (d) at respective time points.

**Figure 8 ijms-23-00056-f008:**
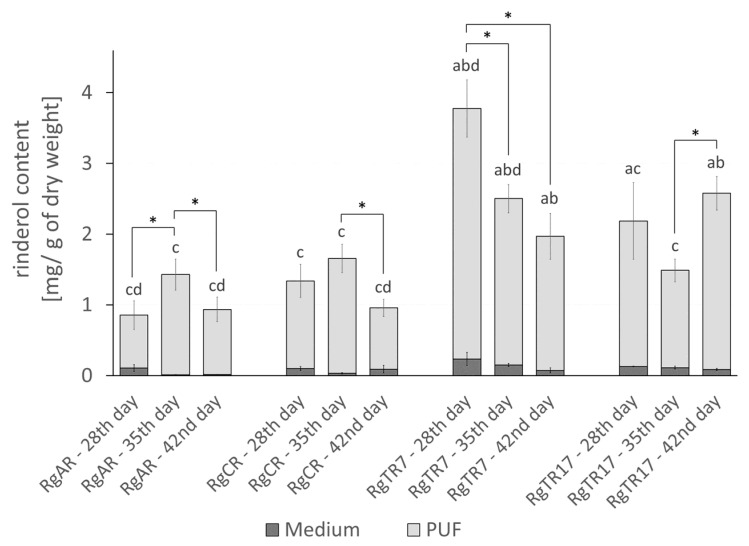
Rinderol productivity [mg/g DW] in in vitro cultures of *R. graeca* roots immobilized on PUF. Rinderol was detected only in post-culture medium and PUF, while it was not present in the root biomass. Presented data are the means ± SD of three independent replicates. Asterisks (*) indicate statistically significant differences (*p* ≤ 0.05) in rinderol productivity between consecutive time points within each root line. Letters indicate statistically significant differences (*p* ≤ 0.05) in rinderol productivity between root lines: RgAR (a), RgCR (b), RgTR7 (c), and RgTR17 (d) at respective time points.

**Figure 9 ijms-23-00056-f009:**
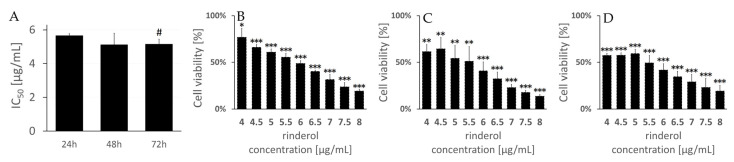
Rinderol cytotoxicity against HeLa cervical cancer cells after 24, 48, and 72 h of incubation obtained in MTT assay. (**A**) Summarized results of IC_50_ for investigated incubation periods with rinderol. Grid (#) indicates a statistically significant difference compared to IC_50_ value for 24 h incubation period (*p* ≤ 0.05). (**B**–**D**) Dose-response curves of HeLa cells exposed to rinderol for 24 h (**B**), 48 h (**C**), and 72 h (**D**), expressed as % of control cells. Presented data are the means ± SD of three independent replicates. Asterisks (*) indicate statistically significant difference between respective rinderol concentration and control cells (exposed to solvent control) set as 100% viability (* *p* ≤ 0.05, ** *p* ≤ 0.01, *** *p* ≤ 0.001).

**Figure 10 ijms-23-00056-f010:**
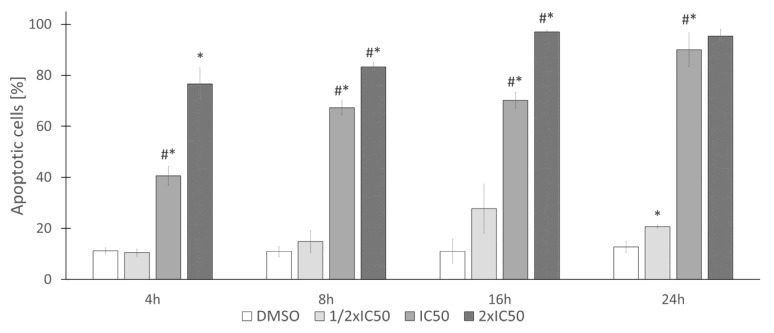
Rinderol proapoptotic activity expressed as a percent of apoptotic HeLa cells (sum of early apoptotic An^+^PI^−^ and late apoptotic An^+^PI^+^ cells) after exposure to studied compound. Data are the means ± SD of at least three independent replicates. Asterisk (*) indicates statistically significant differences (*p* ≤ 0.05) between consecutive rinderol concentrations within the same incubation period (DMSO—0.4%, 1/2 IC_50_—2.83 µg/mL, IC_50_—5.66 µg/mL, 2× IC_50_—11.32 µg/mL). Grids (#) indicate results which were significantly different (*p* ≤ 0.05) in comparison to the previous time point of the same rinderol concentration.

**Figure 11 ijms-23-00056-f011:**
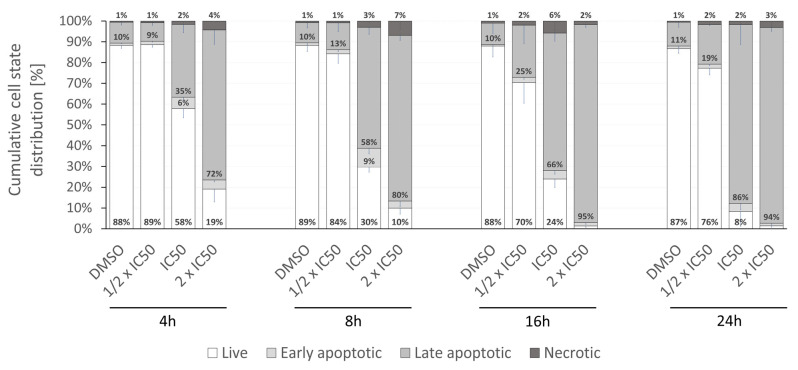
Detailed cell state distribution after HeLa cells exposure to rinderol. Time and dose-dependent shift toward apoptotic cell death are demonstrated and expressed as a percent of HeLa cells (live, early apoptotic, late apoptotic, or necrotic) after exposure to the studied compound. Presented data are the means ± SD of at least three independent replicates. HeLa cells were treated with rinderol in respective concentrations: 1/2 IC_50_–2.83 µg/mL, IC_50_–5.66 µg/mL, 2× IC_50_–11.32 µg/mL, DMSO–0.4%.

**Figure 12 ijms-23-00056-f012:**
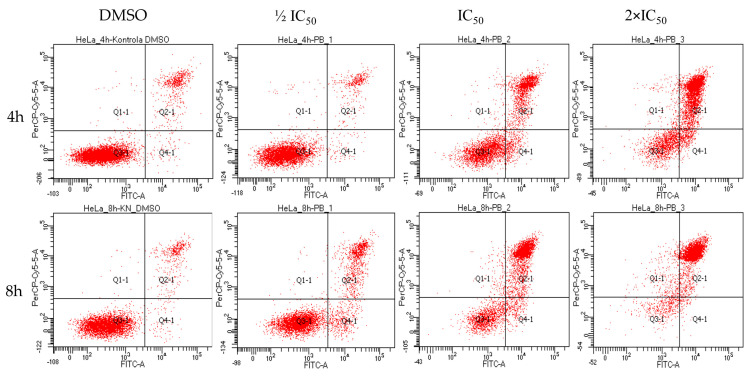
Representative flow cytograms were obtained by double staining Annexin V (FITC) and propidium iodide (PI). Characteristic time and dose-dependent shift of HeLa cells population from An^−^PI^−^ (lower left quadrant) to An^+^PI^+^ (upper right quadrant) progress through An^+^PI^−^ state (lower right quadrant) at each cytogram, respectively. Studied concentrations were related to IC_50_ result of 24 h cytotoxicity assay (1/2 IC_50_–2.83 µg/mL, IC_50_–5.66 µg/mL, 2× IC_50_–11.32 µg/mL, DMSO–0.4%).

**Figure 13 ijms-23-00056-f013:**
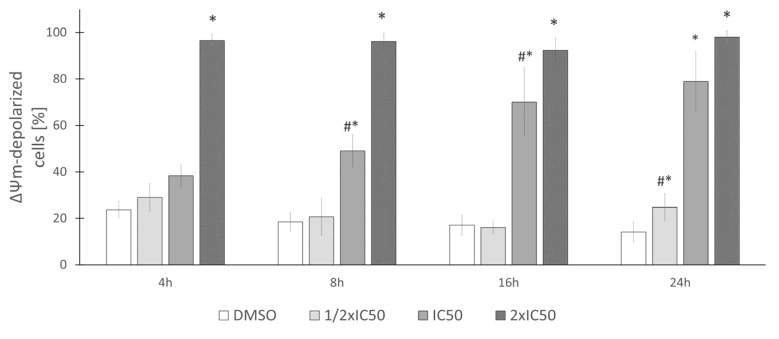
Rinderol induced alterations in mitochondrial membrane potential expressed as a percent of HeLa cell population with depolarized mitochondrial membrane. Data are the means ± SD of at least three independent replicates. Asterisk (*) indicates statistically significant differences (*p* ≤ 0.05) between consecutive rinderol concentrations within the same incubation period (DMSO–0.4%, 1/2 IC_50_–2.83 µg/mL, IC_50_–5.66 µg/mL, 2× IC_50_–11.32 µg/mL). Grids (#) indicate results which were significantly different (*p* ≤ 0.05) in comparison to the previous time point of the same rinderol concentration.

**Figure 14 ijms-23-00056-f014:**
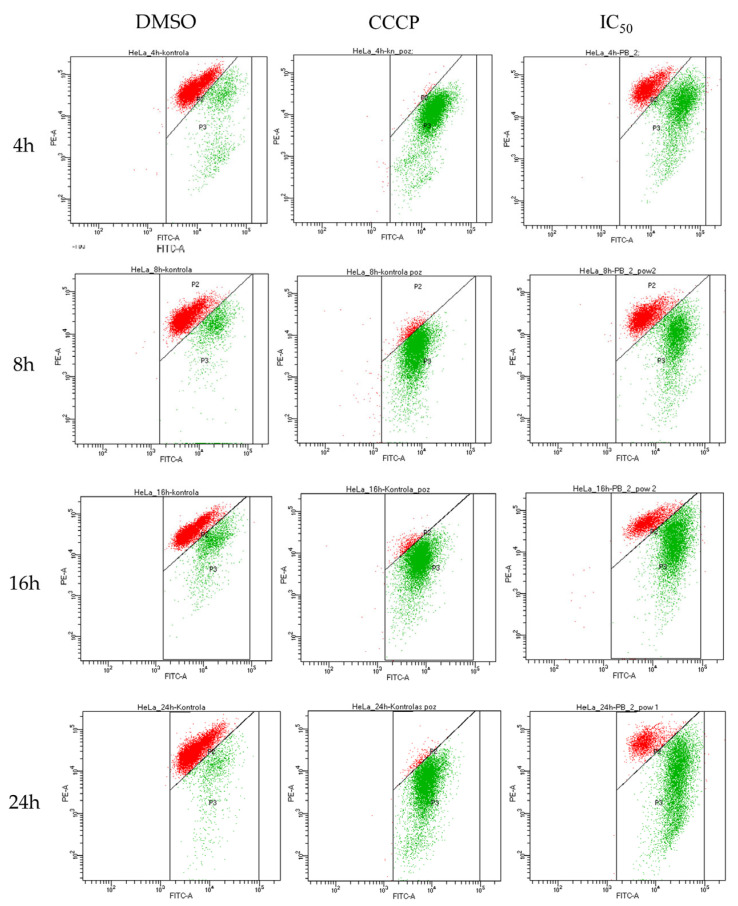
Representative flow cytograms of JC-1 staining demonstrating rinderol-induced mitochondrial membrane depolarization. Characteristic time-dependent shift of fluorescence representing the potential of mitochondrial membrane was observed. CCCP was applied as a positive control. In the presented results, the studied concentration related to IC_50_ result of 24 h cytotoxicity assay (IC_50_–5.66 µg/mL, DMSO–0.4%, CCCP–10 µg/mL) is demonstrated.

**Figure 15 ijms-23-00056-f015:**
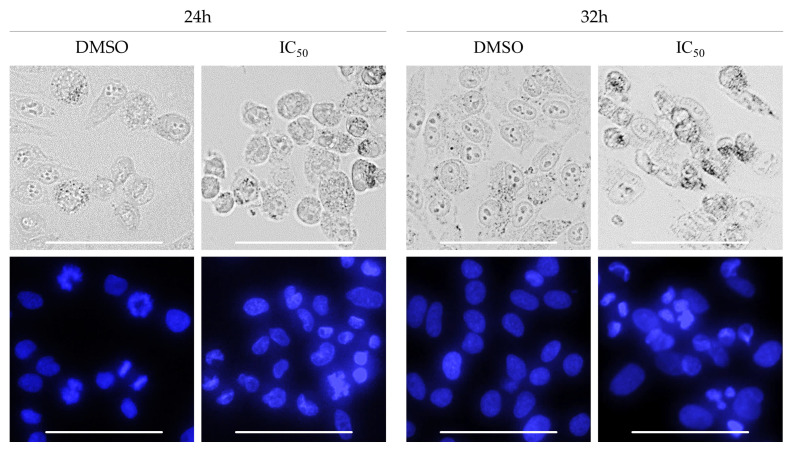
Morphological analysis of HeLa cells stained with DAPI fluorescent dye incubated with or without rinderol. Features of advanced apoptotic cell death (chromatin condensation and occurrence of apoptotic bodies) were observed after 24 h of incubation with investigated compound in 24 h IC_50_ concentration (5.66 µg/mL, DMSO–0.4%). Scale bar: 100 µm.

## Data Availability

The data that support the findings of this study are available from the corresponding author upon reasonable request.

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
