# Peer review of "Polyurethane Foam Rafts Supported In Vitro Cultures of Rindera graeca Roots for Enhanced Production of Rinderol, Potent Proapoptotic Naphthoquinone Compound"

_ijms, 2021, doi:10.3390/ijms23010056_

Round 1
Reviewer 1 Report
The manuscript titled "Polyurethane foam rafts supported in vitro cultures of Rindera graeca roots for enhanced production of rinderol, potent proapoptotic naphthoquinone compound” provides a new insight in the potential production of rinderol in in vitro cultures as well as the pharmacological potential of this phytochemical. The experimental design was well planned, and the work presents high originality. However, some points require further clarification.
In order to improve the manuscript, the authors should address some critical points:
- English language revision and spell check are recommended (e.g. “in vitro”, “in situ” in italic; “IC50” the numbers below the line). Sentences such as those presented in, for example, lines 42-45, line 345, or line 293, should be revised.
- Figures 2, 3, and 4 quality should be improved. Instead of using the acquisition software layout, the authors could improve the overall quality of the presentation of the chromatograms. In addition, rinderol’s chromatogram and UV/Vis spectra should be added, as it was the standard used for identification and quantification.
- In line 105, the authors refer to “Extracts derived from in vitro cultivated roots, post-culture media, and PUF rafts” and provide information regarding TLC separation. This information should be moved to the methodologies section. In lines 181-183 is mentioned an “in-depth analysis of the shares of each of culture system phases (roots biomass, culture medium, and PUF rafts” and section 4.2 mentions extracts from post- culture media and PUF rafts. Section 4.3 states that “Extracts obtained from all types of conducted R. graeca root cultures were collected and combined for preparative isolation with further purification of rinderol”. Then, rinderol content is expressed per flask in both PUF rafts and medium, although the horizontal axis caption always mentions “PUF” (Figure 7) and the content per dry weight (Figure 8) is presented only for PUF rafts. Figures 2, 3, and 4 present chromatograms for rinderol presence in root biomass, media and PUF. The methodologies section should be modified to clearly indicate which fractions were combined and/or analyzed for rinderol content and the results uniformized to represent all the fractions analyzed.
- The expression “phytochemical analysis” should be removed, as the authors analyzed the presence of a single compound and not a screening of the various phytochemicals present.
- In figure 5 the authors present the different effects of PUF root immobilization in biomass growth depending on the root lines. Giving the results obtained, why did the authors present rinderol content as µg/flask and mg/g DW (Figures 7 and 8)? The different inhibitions of PUF rafts on biomass growth are normalized when expressed in dry weight, please justify the necessity of both figures.
- Please add the meaning of “con” and “PUF” to figure 5 caption.
- In figure 5 and figure 7, letters of statistical significant are missing for control groups and RgAR, respectively. In Figure 8 the first two groups are also missing this statistical significance with letter code.
- Please consider the addition of figures concerning the dose-response curves of HeLa cells exposed to rinderol that were used to calculate the IC50 values presented in figure 9. Please revise the statistical analysis as the IC50 for 48h does not appear to be significantly different.
- In order to improve the quality of the data presented, the authors should consider the reformulation of figure 10. Cell distribution in healthy cells, early apoptosis, late apoptosis, and necrosis would benefit the manuscript. In addition to the caption “1/2 IC50, IC50, 2x IC50”, figure 10 caption should provide the corresponding rinderol concentration, as well as the DMSO concentration used in the control.
- Figure 12 should include flow cytometry plots for JC-1 staining and fluorescence shift, such as presented in figure 11 for annexin/PI staining. A plot for control, CCCP, and rinderol should be provided.
11. In line 299, the authors could use “suggest” instead of “indicate”. Although various hallmarks of the apoptotic process were confirmed after rinderol exposure, other known cell death mechanisms can induce part of these responses. As stated by the authors later in the text, the study of proapoptotic proteins is highly relevant to confirm the rinderol-induced apoptosis.
- The conclusion section was added twice (section 3 and section 5).
- In section 4.2, why did the authors use two different solvents for the extraction? Did hexane present a limitation in rinderol extraction from PUF? Was the extraction optimized to guarantee 100% of rinderol extraction regardless of the solvent?
- In section 4.3, please complete the elution program used in HPLC-DAD analysis (run time, temperature, and injection volume). Was the equipment coupled to a column compartment or was the analysis performed at room temperature?
- In section 4.4, the expression “used for all cell cultures” should be revised, as only one cell culture was used.
- Section 4.9 should be moved to before the flow cytometry assays. Additionally, information regarding the number of gated events in each assay is missing.
- In section 4.8, the authors mention the use of sulforhodamine 101, a fluorescent probe not mentioned or presented in the results section. Please revise.
Author Response
Dear Reviewer,
we would like to kindly express our gratitude for the whole effort put into the process of manuscript review and comprehensive guidance regarding to necessary revisions. Received expert opinion as the best example of constructive criticism was invaluable help and without a doubt allowed to increase quality of submitted paper. Moreover, we want to thank for the opportunity to introduce corrections in ongoing reviewing process. Along with our gratitude we would like to provide improved manuscript prepared thoroughly with consideration of all Reviewers suggestions. All corrections in the main text of the manuscript were marked and for the full clarity of the process we would like to provide point by point response to Reviewer’s comments:
- English language revision and spell check are recommended (e.g. “in vitro”, “in situ” in italic; “IC50” the numbers below the line). Sentences such as those presented in, for example, lines 42-45, line 345, or line 293, should be revised.
Revision of English language and spell check were performed as indicated. Also the whole text of the manuscript was once again double checked and necessary corrections were introduced in “track changes” mode.
- Figures 2, 3, and 4 quality should be improved. Instead of using the acquisition software layout, the authors could improve the overall quality of the presentation of the chromatograms. In addition, rinderol’s chromatogram and UV/Vis spectra should be added, as it was the standard used for identification and quantification.
Presentation of chromatography results has been modified aiming to improve quality of chromatograms and the clarity. Most of the data was transferred to Supplementary materials where we were able to hold all chromatograms as enlarged images without the negative impact on the manuscript reader experience. For the main text we selected only the data relating to most productive culture system (transformed root line 7 – TR7, Figures 2 and 3) and for the rinderol identification standard (Figure 4).
- In line 105, the authors refer to “Extracts derived from in vitro cultivated roots, post-culture media, and PUF rafts” and provide information regarding TLC separation. This information should be moved to the methodologies section.
According to the suggestion we moved indicated fragment to Materials and Methods chapter, more specifically to section 3.4. titled “Isolation of rinderol and HPLC-DAD-UV-Vis analysis”.
In lines 181-183 is mentioned an “in-depth analysis of the shares of each of culture system phases (roots biomass, culture medium, and PUF rafts” and section 4.2 mentions extracts from post- culture media and PUF rafts. Section 4.3 states that “Extracts obtained from all types of conducted R. graeca root cultures were collected and combined for preparative isolation with further purification of rinderol”.
During experiment we analyzed the content of rinderol in all in vitro culture phases aiming to reveal main place of its accumulation. Extraction procedure included dry root biomass, post-culture medium and PUF-cuboids as it was described in Materials and Methods (in previous section 4.3 – now renumbered to section 3.3 and 3.4). However HPLC analysis demonstrated lack of rinderol in root biomass after the culture. Almost all amount accumulated in PUF and small fraction in culture medium, being the reason of not incorporating the root biomass into the productivity plot. In order to avoid this lack of clarity in future we introduced corrections to the plot, caption and main text.
Then, rinderol content is expressed per flask in both PUF rafts and medium, although the horizontal axis caption always mentions “PUF” (Figure 7) and the content per dry weight (Figure 8) is presented only for PUF rafts.
Horizontal axis captions “PUF” were replaced to avoid possible confusion for the reader.
Figures 2, 3, and 4 present chromatograms for rinderol presence in root biomass, media and PUF.
In line with annotation 2, we revised chromatograms attached to the main text of the manuscript. We selected the most representative plots with related spectrum (most productive culture system) and resized it with better quality for the viewer and the rest of chromatograms was transferred to the Supplementary materials hoping to preserve clarity of the manuscript. In presented study the experimental culture systems immobilized on PUF-supports resulted in the lack of compound detection in root biomass, probably due to efficient in situ product removal and accumulation of rinderol in PUF. It is a reason why in the main text chromatograms of root biomass are not attached, however can be easily incorporated if necessary.
The methodologies section should be modified to clearly indicate which fractions were combined and/or analyzed for rinderol content and the results uniformized to represent all the fractions analyzed.
The suggested modification was performed and we hope that introduced text modifications in methodology section will be sufficient for satisfactory clarity of performed and described experimental procedures.
- The expression “phytochemical analysis” should be removed, as the authors analyzed the presence of a single compound and not a screening of the various phytochemicals present.
Expression was revised in line with above indication and now expression “chromatographic analysis” is used.
- In figure 5 the authors present the different effects of PUF root immobilization in biomass growth depending on the root lines. Giving the results obtained, why did the authors present rinderol content as µg/flask and mg/g DW (Figures 7 and 8)? The different inhibitions of PUF rafts on biomass growth are normalized when expressed in dry weight, please justify the necessity of both figures.
Presentation of both figures and resulting expression of rinderol productivity in both units was submitted with purpose to provide more detailed productivity characteristics of investigated immobilized culture systems. Despite the µg/flask unit seems to be more suited to express yield of culture system especially in terms of potential scalability, the mg/g of DW unit allows to compare obtained results to the metabolites content in conventionally field harvested biomass or ground-cultivated plants.
- Please add the meaning of “con” and “PUF” to figure 5 caption.
Indicated abbreviations from Figure 5 were introduced and described in caption under the plot.
- In figure 5 and figure 7, letters of statistical significant are missing for control groups and RgAR, respectively. In Figure 8 the first two groups are also missing this statistical significance with letter code.
All indicated plots were modified according to suggestions. Symbols of statistical significance were added in missing places and statistical analysis was once again checked.
- Please consider the addition of figures concerning the dose-response curves of HeLa cells exposed to rinderol that were used to calculate the IC50 values presented in figure 9. Please revise the statistical analysis as the IC50 for 48h does not appear to be significantly different.
Dose response curves of MTT assay were added to the Figure 9.
- In order to improve the quality of the data presented, the authors should consider the reformulation of figure 10. Cell distribution in healthy cells, early apoptosis, late apoptosis, and necrosis would benefit the manuscript. In addition to the caption “1/2 IC50, IC50, 2x IC50”, figure 10 caption should provide the corresponding rinderol concentration, as well as the DMSO concentration used in the control.
In response to above suggestion we designed additional plot fulfilling the aim to visualize specific cell state distribution in function of time and rinderol concentration (Figure 11). We would like to propose to also supplementary leave the primary, simplified plot with only apoptotic cells, as it allows for more clarity in description of statistical relations between experimental time and dose dependent effect of rinderol activity.
- Figure 12 should include flow cytometry plots for JC-1 staining and fluorescence shift, such as presented in figure 11 for annexin/PI staining. A plot for control, CCCP, and rinderol should be provided.
Along with provided suggestions we designed and attached the cytometry plots for JC-1 assay (Figure 14) aiming to demonstrate fluorescence shift caused by rinderol exposition.
- In line 299, the authors could use “suggest” instead of “indicate”. Although various hallmarks of the apoptotic process were confirmed after rinderol exposure, other known cell death mechanisms can induce part of these responses. As stated by the authors later in the text, the study of proapoptotic proteins is highly relevant to confirm the rinderol-induced apoptosis.
Correction was performed as indicated.
- The conclusion section was added twice (section 3 and section 5).
Section 3 was removed.
- In section 4.2, why did the authors use two different solvents for the extraction? Did hexane present a limitation in rinderol extraction from PUF? Was the extraction optimized to guarantee 100% of rinderol extraction regardless of the solvent?
Application of methanol for rinderol extraction from PUF was caused by better permeability of selected solvent with lesser effect on the PUF structure what resulted in higher isolation effectivity without potential contamination. Due to pigment nature of rinderol causing strong coloration all extractions were performed until the solvent color fade what correlated well with HPLC analysis. Standard protocol of n-hexane extraction from post-culture medium and roots allows for better selectivity of the procedure toward rinderol which is a naphthoquinone derivative.
- In section 4.3, please complete the elution program used in HPLC-DAD analysis (run time, temperature, and injection volume). Was the equipment coupled to a column compartment or was the analysis performed at room temperature?
All necessary data was added in section 4.3. (renumbered to 3.4.) according to suggestions.
- In section 4.4, the expression “used for all cell cultures” should be revised, as only one cell culture was used.
Suggested revision was performed.
- Section 4.9 should be moved to before the flow cytometry assays. Additionally, information regarding the number of gated events in each assay is missing.
Section 4.9 has been moved and renumbered to 3.7 according to corrected ordering.
- In section 4.8, the authors mention the use of sulforhodamine 101, a fluorescent probe not mentioned or presented in the results section. Please revise.
Sentence has been revised accordingly. Applied solution contained sulforhodamine however its visualization was not the objective of experiment.
Reviewer 2 Report
In this interesting work, the authors present the successful application of in vitro root cultures of R. graeca roots, immobilized on polyurethane foam rafts for production of rinderol, a novel lipophilic furanonaphthoquinone. Preliminary screening demonstrated rinderol cytotoxicity against various human cancer cell lines.
This biotechnological approach for sustainable production of natural compound can be an important result in view of the growing employment on large scale applications of secondary metabolites.
Anyway, I suggest some recommendations
Introduction
Line 82 The presence of apoptosis and mitochondrial membrane alterations are not sufficient to elucidate mechanism of cytotoxicity. Please, soften the sentence
Results and Discussion
Please, go through the ms and check the presentation of the results of the statistical analyses relevant to your study, especially in figure 8, 10 and 12
For example, “The highest root biomass accumulation was achieved by roots of RgCR line after 42 days of growth in control culture”: the statistical difference in figure 5 between RgCR con 42, RgARcon42, RgTR7 con42 and RgTR17 con 42 is not immediately clear to the reader, their SD bars are even overlapping, thus their weights could be statistically not different…
In the caption of figure 10 is written: Asterisk (*) indicates statistically significant difference between control and rinderol-treated cells (within groups, I suppose), but these asterisks do not explain Line 250 “Analysis of obtained results suggests clearly time and dose dependent cell death inducing effect of rinderol” (between groups, I suppose)
In figure 12, again, for each considered time I note equal ΔΨ , I can't see a time dependent depolarization...
I think that a more detailed presentation of the statistical analysis results would make easier to the readers the text comprehension
2.1. HPLC DAD analysis of Rindera graeca in vitro root cultures
Line 86 The suggestion of the production of naphthoquinones specific for Boraginaceae plants family coming from of a visual analysis of the of post-culture medium colour is quite hard to argument. I suggest deleting the sentence.
In my version of the ms it’s hard to understand what is written inside the panels of the chromatograms
2.4. Effect of root immobilization on rinderol production
I would add a 2.5 paragraph including the results regarding the biological activity of rinderol
Rinderol cytotoxicity against HeLa cervical cancer cells (fig 9): solvent control is missing. Please add in the caption the rinderol concentration
Fig 5: explain in the caption the term “con” used in the axis
Fig 10: add rinderol concentration in the caption
Have you compared the rinderol cytotoxicity with a positive control?
- Materials and Methods
Line 353: the growth rate was expressed as a ratio of final dry weight to initial dry weight. I can’t find in the text the growth rate results
Please, detail the extraction protocol and the extract drying procedure
Why did you choose different solvents for roots/post culture medium and PUF (hexane and methanol, respectively)?
In the text some typing errors are spread
Author Response
Dear Reviewer,
we would like to kindly express our gratitude for the whole effort put into the process of manuscript review and comprehensive guidance regarding to necessary revisions. Received expert opinion as the best example of constructive criticism was invaluable help, and without a doubt allowed us to increase quality of submitted paper. Moreover, we want to thank for the opportunity to introduce corrections in ongoing reviewing process. Along with our gratitude we would like to provide improved manuscript prepared thoroughly with consideration of all Reviewers suggestions. All corrections in the main text of the manuscript were marked and for the full clarity of the process we would like to provide point by point response to Reviewer’s comments:
Introduction
Line 82 The presence of apoptosis and mitochondrial membrane alterations are not sufficient to elucidate mechanism of cytotoxicity. Please, soften the sentence
Indicated sentence was revised according to above suggestions.
Results and Discussion
Please, go through the ms and check the presentation of the results of the statistical analyses relevant to your study, especially in figure 8, 10 and 12
For example, “The highest root biomass accumulation was achieved by roots of RgCR line after 42 days of growth in control culture”: the statistical difference in figure 5 between RgCR con 42, RgARcon42, RgTR7 con42 and RgTR17 con 42 is not immediately clear to the reader, their SD bars are even overlapping, thus their weights could be statistically not different…
In the caption of figure 10 is written: Asterisk (*) indicates statistically significant difference between control and rinderol-treated cells (within groups, I suppose), but these asterisks do not explain Line 250 “Analysis of obtained results suggests clearly time and dose dependent cell death inducing effect of rinderol” (between groups, I suppose)
In figure 12, again, for each considered time I note equal ΔΨ , I can't see a time dependent depolarization...
I think that a more detailed presentation of the statistical analysis results would make easier to the readers the text comprehension
In response to above suggestions all plots introduced to manuscript were revised regarding to statistical data visualization and comprehensive modifications were introduced. We hope that presented coverage will be satisfactory and all statistically significant relations are clearly arranged.
2.1. HPLC DAD analysis of Rindera graeca in vitro root cultures
Line 86 The suggestion of the production of naphthoquinones specific for Boraginaceae plants family coming from of a visual analysis of the of post-culture medium colour is quite hard to argument. I suggest deleting the sentence.
Pointed sentence was deleted.
In my version of the ms it’s hard to understand what is written inside the panels of the chromatograms
Chromatograms were resized and partially transferred to Supplementary materials. In main text we would like to propose leaving the plots for exemplary, representative samples from immobilized TR7 root line which was the most productive culture system in our experiment. Moreover we would like to add chromatogram of rinderol standard with its spectrum which was used for identification and quantifications.
2.4. Effect of root immobilization on rinderol production
I would add a 2.5 paragraph including the results regarding the biological activity of rinderol
Additional paragraph was added.
Rinderol cytotoxicity against HeLa cervical cancer cells (fig 9): solvent control is missing. Please add in the caption the rinderol concentration
Fig 5: explain in the caption the term “con” used in the axis
Abbreviations description was added to the caption.
Fig 10: add rinderol concentration in the caption
Rinderol concentrations were added to the figure caption.
Have you compared the rinderol cytotoxicity with a positive control?
In the current cytotoxicity experiments rinderol was not yet compared with positive controls.
- Materials and Methods
Line 353: the growth rate was expressed as a ratio of final dry weight to initial dry weight. I can’t find in the text the growth rate results
Growth rate results are provided together with final weights in common parentheses.
Please, detail the extraction protocol and the extract drying procedure
To provide detailed description of extraction and drying procedures we slightly modified Materials and methods section creating additional subsection 3.3 devoted to “Extraction procedure” and providing additional information about our experimental methods.
Why did you choose different solvents for roots/post culture medium and PUF (hexane and methanol, respectively)?
Application of methanol for rinderol extraction from PUF was caused by better permeability of selected solvent with lesser effect on the PUF structure what resulted in higher isolation effectivity without potential contamination. Due to pigment nature of rinderol causing strong coloration all extractions were performed until the solvent color fade what correlated well with HPLC analysis. Standard protocol of n-hexane extraction from post-culture medium and roots allows for better selectivity of the procedure toward rinderol which is a naphthoquinone derivative.
In the text some typing errors are spread.
The manuscript was checked to remove typing errors.

Round 2
Reviewer 1 Report
Regarding the article titled " Polyurethane foam rafts supported in vitro cultures of Rindera graeca roots for enhanced production of rinderol, potent proapoptotic naphthoquinone compound", the authors have carefully followed the reviewers' suggestions, namely in text and figures editing. The supplementary material and newly added figures greatly increase the manuscript quality. However, some points remain unaddressed:
- English language and grammar revision is recommended (e.g. "Technological ascendancy of plants"; "CH2Cl2/EtOAc" underline numbers and present the full name when they appear in the text for the first time;
- The chromatograms presented in the supplementary material display and higher quality than the ones in the manuscript. Please edit the figures to remove the acquisition software layout and better show the chromatograms and spectra, as the authors have done in the supplementary material;
- In figures 5, 7 and 8, please revise the new statistics. To facilitate the reader's comprehension, the letters and asterisk should maintain their meaning through the various figures. For example in Figure 5, if the letters represent differences within the root line (3 points) why does it vary from A to L? In figure 7 and 8 statistics symbology can be confusing, in this case, are the grids necessary if all consecutive time points are significantly different and therefore present and "*"? Please revise and simplify;
- Figure 9 (panels B, C, and D) are missing statistical treatment;
- In figure 11 caption, please also add the respective rinderol concentration as in the previous caption (e.g. DMSO; IC50, 2x IC50). Uniformize all captions with this information;
- Flow cytometry assay still misses the number of gated events.
Reviewer 2 Report
The authors have followed the reviewers' suggestions and increased the manuscript quality. However, some minor points are to be clarified.
- Statistical analysis (fig 5, 7, 8,13)
Are you sure that you can’t find an easier way for the readers to understand your results?
Please add the rinderol concentrations in all captions (see figure 11)
The full names should be presented when they appear in the text for the first time.
The chromatograms in supplementary material are now good, please adopt the same layout for the ones in the text
